# Neuronal and Astrocytic Extracellular Vesicle Biomarkers in Blood Reflect Brain Pathology in Mouse Models of Alzheimer’s Disease

**DOI:** 10.3390/cells10050993

**Published:** 2021-04-23

**Authors:** Francheska Delgado-Peraza, Carlos J. Nogueras-Ortiz, Olga Volpert, Dong Liu, Edward J. Goetzl, Mark P. Mattson, Nigel H. Greig, Erez Eitan, Dimitrios Kapogiannis

**Affiliations:** 1Laboratory of Clinical Investigation, Intramural Research Program, National Institute on Aging, National Institutes of Health, Baltimore, MD 212241, USA; francheska.delgado-peraza@nih.gov (F.D.-P.); carlos.nogueras-ortiz@nih.gov (C.J.N.-O.); 2NeuroDex Inc., Natick, MA 01760, USA; ovolpert@neurodex.co (O.V.); eeitan@neurodex.co (E.E.); 3Translational Gerontology Branch, Intramural Research Program, National Institute on Aging, National Institutes of Health, Baltimore, MD 21224, USA; liudo@grc.nia.nih.gov (D.L.); greign@grc.nia.nih.gov (N.H.G.); 4Department of Medicine, University of California, San Francisco, CA 94143, USA; edward.goetzl@ucsf.edu; 5San Francisco Campus for Jewish Living, San Francisco, CA 94112, USA; 6Department of Neuroscience, Johns Hopkins School of Medicine, Baltimore, MD 21205, USA; mmattso2@jhmi.edu

**Keywords:** extracellular vesicles, exosomes, Alzheimer’s, transgenic, biomarkers, Tau, beta-amyloid, complement

## Abstract

Circulating neuronal extracellular vesicles (NEVs) of Alzheimer’s disease (AD) patients show high Tau and β-amyloid (Aβ) levels, whereas their astrocytic EVs (AEVs) contain high complement levels. To validate EV proteins as AD biomarkers, we immunocaptured NEVs and AEVs from plasma collected from fifteen wild type (WT), four 2xTg-AD, nine 5xFAD, and fifteen 3xTg-AD mice and assessed biomarker relationships with brain tissue levels. NEVs from 3xTg-AD mice had higher total Tau (*p* = 0.03) and p181-Tau (*p* = 0.0004) compared to WT mice. There were moderately strong correlations between biomarkers in NEVs and cerebral cortex and hippocampus (total Tau: cortex, r = 0.4, *p* = 0.009; p181-Tau: cortex, r = 0.7, *p* < 0.0001; hippocampus, r = 0.6, *p* < 0.0001). NEVs from 5xFAD compared to other mice had higher Aβ42 (*p* < 0.005). NEV Aβ42 had moderately strong correlations with Aβ42 in cortex (r = 0.6, *p* = 0.001) and hippocampus (r = 0.7, *p* < 0.0001). AEV C1q was elevated in 3xTg-AD compared to WT mice (*p* = 0.005); AEV C1q had moderate-strong correlations with C1q in cortex (r = 0.9, *p* < 0.0001) and hippocampus (r = 0.7, *p* < 0.0001). Biomarkers in circulating NEVs and AEVs reflect their brain levels across multiple AD mouse models supporting their potential use as a “liquid biopsy” for neurological disorders.

## 1. Introduction

Alzheimer’s disease (AD) is neuropathologically characterized by the aggregation of β-amyloid (Aβ) into plaques extracellularly and hyperphosphorylated Tau into neurofibrillary tangles within neurons [1,2]. The current biological definition of AD is based on brain accumulation of Aβ (amyloidosis) and Tau (tauopathy) opening the way for its diagnosis based on biomarkers [3]. Currently, widely accepted biomarkers for AD are derived from cerebrospinal fluid (CSF) analysis or employ positron emission tomography (PET), limiting their broad clinical implementation due to their invasiveness and high costs. Therefore, there is an urgent need for blood-based biomarkers for preclinical and clinical diagnosis of AD, evaluation of disease progression, and as evidence of target engagement by experimental treatments [4]. 

Extracellular vesicles (EVs) isolated from peripheral blood are emerging as a source of biomarkers for neurodegenerative diseases [5]. EVs are membranous nanoparticles ranging in size from ~50 to 1000 nm that are released by all cell types and can be isolated from all biofluids [6]. Circulating EVs are a mixture of many vesicle types, mainly exosomes derived from late endosomes and microvesicles budding from the plasma membrane. Collectively, circulating EVs reflect the molecular composition of their cells of origin [7]. Multiple studies have shown neuron- and astrocyte-specific cargo in blood EVs [8,9,10] and produced evidence that EVs released by brain cells can cross the blood-brain barrier [11]. Consequently, we and others have developed methodologies for immunocapturing neuronal origin-enriched EVs (NEVs) from peripheral blood using antibodies against L1 Cell Adhesion Molecule (L1CAM) and/or other neuronal markers [8,9,10,12,13,14], and the glutamate aspartate transporter-1 (GLAST) for astrocytic origin-enriched EVs (AEVs) [15,16]. NEVs and AEVs isolated from human plasma show a multi-fold enrichment in neuron- and astrocyte-specific cargos [9,10,17,18], respectively.

NEVs and AEVs from AD patients at the clinical and preclinical stages carry higher levels of AD pathogenic proteins compared to controls, including total Tau (tTau), phosphorylated Thr181 Tau (p181-Tau), and Aβ42 in NEVs [9,10,12,14,16,17,19], and complement proteins, including C1q, in AEVs [15]. Importantly, AD pathology is complex and variable between patients. Stratification of patients based on specific pathologies is the basis of Precision Medicine. Although the presence of pathology markers in NEVs and AEVs suggests these platforms may identify brain pathology in AD, the relationship of EV biomarkers and brain pathology has not been directly demonstrated. To address this knowledge gap, we measured tTau, p181-Tau, and Aβ42 in NEVs and C1q in AEVs, as well as matched brain tissue samples from wild-type (WT), and 2xTg-AD, 3xTg-AD and 5xFAD mice. These represent diverse mice models of AD, which have been extensively characterized and accepted as models for AD [20,21,22,23,24,25,26,27,28,29]. The 2xTg-AD amyloidosis mouse model expresses the human amyloid precursor protein (APP) KM670/671NL (Swedish; APP_swe_) and presenilin-1 (PSEN1) ΔE9 mutations in the central nervous system (CNS) under the control of the mouse prion protein promoter [28], resulting in pathological APP processing and accumulation of Aβ plaques. The 5xFAD is a newer model that recapitulates Aβ pathology more rapidly by overexpressing human APP and PSEN1 transgenes with a total of five mutations (APP_swe_, APP I716V (Florida), APP V717I (London), PSEN1 M146L (A > C), and PSEN1 L286V) [29]. 3xTg-AD mice exhibit both Aβ plaque and Tau tangle pathologies, by overexpressing human APP_swe_, and Tau P301L in PSEN1 M146V mutant knockin mice [25]. Astrocytic complement expression is elevated in mouse models of tauopathy and amyloidosis. A striking C1q protein level increase in the P301S mouse model of tauopathy is associated with synaptic degeneration [30]. The different mice represent slightly different aspects and severity of the disease and, thus, allow the evaluation of the concordance between plasma EVs and the CNS tissue.

Higher levels of tTau, p181-Tau and C1q were observed in the 3xTg mice cortex and hippocampus, which were also reflected in their EVs. Similarly, higher levels of Aβ42 were observed in the 5XFAD mice cortex and hippocampus, which were also reflected in their EVs. The clear separation between different AD mice models and the significant concordance between the tissue and the EVs provide further support to the use of circulating EVs as a form of “liquid biopsy” for AD precision medicine. 

## 2. Materials and Methods

### 2.1. Mice

We analyzed a total of 43 mice; 29 female, 14 male; age: 9.6 ± 2.4 (5–12 months): four 2xTg-AD (all female; age: 8.3 ± 0.12 months), fifteen 3xTg-AD (nine male, six female; age: 9.1 ± 0.8, 6–10.5 months), nine 5xFAD (all female; 12 months old) and 15 wild-type (WT; 5 male, 10 female; age: 8.9 ± 3.2, 5–12 months) mice. Breeding colonies were maintained at the National Institute on Aging (NIA) Intramural Research Program vivarium. Mice were housed 3–4 per cage with ad libitum access to food and water in a room with a reverse 12-h light/12-h dark cycle, with lights on at 18:00 h. All procedures were approved by the NIA Animal Care and Use Committee and complied with National Institutes of Health guidelines (Animal Use Protocol Number: 263-LCI-2020). 

### 2.2. Plasma and Brain Sample Collection and Processing

To acquire enough volume for EV isolation and protein quantification, serial blood samples were collected retro-orbitally every week for five weeks in EDTA polypropylene tubes and centrifuged at 1500× *g* for 10 min at 4 °C within 1 h of collection, supernatant plasma was stored at −80 °C, according to guidelines regarding pre-analytical factors for EV biomarkers [31,32]. Euthanasia was performed three weeks after the last blood collection to give the mice time to fully recover. Mice were anesthetized with isoflurane gas and perfused transcardially with phosphate buffered saline (PBS). Brains were extracted and hemispheres separated. The right hemisphere was placed in 4% paraformaldehyde (PFA) and stored at 4 °C for immunohistochemistry. The cortex (Ctx) and hippocampus (Hp) were dissected from the left hemisphere and homogenized in five times its weight of buffer A (20 mM Tris base, pH 7.4, 150 mM NaCl, 1 mM EDTA, pH 8, supplemented with protease (cOmpleteTM Protease Inhibitor Cocktail; Millipore Sigma, Burlington, MA, USA) and phosphatase (HaltTM Phosphatase Inhibitor Cocktail; Thermo Fisher Scientific Waltham, MA, USA) inhibitors using a hand-held homogenizer (PRO Scientific Bio-Gen PRO200 Handheld Homogenizer, Oxford, CT, USA). Extracts were centrifuged at 22,000× *g* for 10 min and supernatants were transferred to new tubes and subjected to protein determination (Pierce^TM^ BCA Protein Assay Kit, Thermo Fisher Scientific) and an equal amount of proteins were used for downstream immunoblotting, enzyme-linked immunosorbent assays (ELISAs) and Luminex multiarray assays. 

### 2.3. Isolation of NEVs and AEVs 

Plasma samples were thawed on ice and serial blood collection samples from each mouse were pooled. Plasma samples were defibrinated with 30 min incubation with Thrombin (System Biosciences, Mountainview, CA, USA) at room temperature. NEVs expressing L1CAM/CD171 were isolated using NeuroDex ExoSORT^TM^ proprietary protocol for immunocapture of EVs. AEVs were isolated using immunocapture against astrocytic marker GLAST (Miltenyi Biotec, Auburn, CA, USA). Protease and phosphatase inhibitors were applied to multiple steps, as previously described. NEVs and AEVs were lysed with protein extraction solution and lysates were stored at −80 °C until assays. To ensure that ExoSORT^TM^, which was developed for human samples can work with mice plasma, NeuroDex L1CAM fluorescent exosomes were spiked into human and mice plasma samples and the recovery was determined to be similar between the two species (42 ± 7.31% in mice plasma and 46 ± 11.2% in human plasma). 

### 2.4. EV Characterization

Intact EVs were used for determination of particle concentration and diameter using nanoparticle tracking analysis (NTA) (Nanosight NS500; Malvern, Amesbury, UK). The protein concentration was determined using the Bradford protein assay (Bio-Rad, Hercules, CA, USA). Twenty µg of total protein per NEVs sample, 1:50 diluted plasma and 20 µg of HEK293 cell culture lysate was resolved by sodium dodecyl sulfate polyacrylamide gel electrophoresis (SDS-PAGE) using 4–20% Bis-Tris gels in MOPS SDS running buffer (NuPAGE^®^ Novex^®^ SDS-PAGE system; Thermo Fisher Scientific) and transferred to polyvinylidene fluoride membranes (iBlot^®^ 2 gel transfer system; Thermo Fisher Scientific). Membranes were blocked using the Blocker™ FL Fluorescent Blocking Buffer (Thermo Fisher Scientific) for 1 h at RT and incubated overnight at 4 °C with the following antibodies: CD171 (cat. no. 14-1719-82, eBioscience, San Diego, CA, USA), CD63 (cat. no. 143903, BioLegend, San Diego, CA, USA), CD81 (cat. no. 104905, BioLegend) and CD9 (cat. no. 124807, BioLegend), Calnexin (cat. no. MA3-027, Invitrogen, Carlsbad, CA, USA), GRIA2 (cat. no. MBS716094, MyBioSource, San Diego, CA, USA), FLOT1 (cat. no. AB41927, Abcam, Cambridge, MA, USA) and albumin (cat. no. Ab207327, Abcam). Membranes were washed three times with tris buffer saline supplemented with 0.05% Tween 20 for 5 min and blots were incubated with secondary antibodies: Goat anti-rRabbit IgG (H + L) highly cross-adsorbed secondary antibody, Alexa Fluor Plus 488 and goat anti-mouse IgG (H + L) highly cross-adsorbed secondary antibody, Alexa Fluor Plus 647 (Thermo Fisher Scientific). Antibody excess was washed three times with Tris buffer saline supplemented with 0.05% Tween 20 for 5 min and blots were scanned using the iBright Western blot imaging system (Thermo Fisher Scientific). 

ELISAs for APOA and Albumin (R&D Systems, Minneapolis, MN, USA) were performed for plasma (1:1000 dilution) and NEVs (1:4 dilution) samples according to the manufacturer’s protocol. 

Flow cytometry (FC) for EVs was performed according to established protocol [33], with modifications. Briefly, mice plasma CD171-expressing EVs were captured on magnetic beads according to NeuroDex proprietary protocol. The beads with captured EVs were diluted to 1 µg/mL in blocking buffer (0.5% FCS in PBS), blocked for 30 min at room temperature and 100 µL aliqots used for staining with fluorophore PE-tagged antibodies against CD9 (cat. no. 312106, BioLegend) and CD171 (cat. no. 371603, BioLegend). Mock pulldown beads with IgG for immunocapture were used as control. The staining was analyzed using Accuri C6 flow cytometer (Becton Dickinson, Franklin Lakes, NJ, USA) and FlowJo^TM^ software (Becton Dickinson), with unstained beads as control.

EV isolation under our protocol was confirmed by conducting an EV recovery experiment consisting of spiking NeuroDex flourescent L1CAM EVs to mice and human plasma samples and measuring their recovery following NeuroDex propritery ExoSORT^TM^ procedure using a fluorescent plate reader. NeuroDex flourescent L1CAM EVs were isolated from HEK-293 cells stably overexpressing L1CAM-GFP. 

### 2.5. Quantification of Analytes 

We quantified tTau, p181-Tau, and Aβ42 in NEVs using the MILLIPLEX^®^ MAP Human Amyloid Beta and Tau Panel (cat no. HNABTMAG-68K, EMD Millipore Corporation, Billerica, MA, USA). Plates were read using Luminex^®^ 200™ System and the xPOTENT^®^ acquisition software (Luminex Corporation, Austin, TX, USA). Mouse C1q levels in AEVs were quantified using ELISA (cat. no. LS-F8963, LS Bioscience, Seattle, WA, USA). Concentration was based on the five-parameter logistic regression curve-fit of the absorbances registered by serially diluted standards provided by the manufacturer. 

The total amount of samples per experimental group were equally distributed in two different plates and the signals from each sample normalized to the average of all measurements per plate to account for interplate variability. Samples were assessed in duplicate and the mean coefficients of variation (CV) were 14.32 ± 8.25% for tTau, 17.41 ± 9.22% for p181-Tau, 19.11 ± 9.22% for Aβ42, and 16.08 ± 7.95% for C1q. The limit of detection (LOD) for the Luminex kit was defined by the manufacturer to be 11 pg/mL for tTau, 0.7 pg/mL for p181-Tau, and 3 pg/mL for Aβ42. The LOD for the C1q ELISA was calculated as the mean of the blank plus 2.5 the standard deviation (SD) of the blank, to be 0.621 ng/mL for C1q. Samples with signals bellow the LOD but CV < 20% were assigned a value of 0 (Aβ42 in NEVs: 4 WT, 1 2xTg-AD and 5 3xTg-AD samples; C1q in Ctx: 4 WT and 3 3xTg-AD and 5xFAD samples; C1q in Hp: 5 WT, 2 3xTg-AD and 1 5xFAD; C1q in AEVs: 2 WT and 1 3xTg-AD samples). 

### 2.6. tTau and p181-Tau Immunoblotting and Immunohistochemistry 

Immunoblotting was used to confirm the previously described age-dependent accumulation of tTau and p181-Tau in the brain of 3xTg-AD mice showing mild to robust deposition of tTau and p181-Tau from 6 to 12 months of age [25,34]. Thirty micrograms of total protein from brain lysates of 2–6 mice per age group (6, 8 and 10 months old) were used for tTau and p181-Tau quantification. Membranes were incubated overnight at 4 °C with fluorescently labelled antibodies targeting total human Tau (HT7, 1:500 dilution, cat no. MN1000), phosphorylated Tau at Thr181 (AT270, 1:500 dilution, cat no. MN1050) (both from Thermo Fisher Scientific), and β-actin (1:1000 dilution, cat. no. ab8226, Abcam) used as loading control. The optical density of protein bands was quantified using Image Studio^TM^ Lite Software (LI-COR Biosciences, Lincoln, NE, USA) and tTau and p181-Tau levels were normalized to β-actin.

Immunoblotting results for p181-Tau were validated by immunofluorescence histochemistry using a previously described method [35,36] with slight modifications. Briefly, right hemispheres were placed in 4% PFA for 48 h at 4 °C and cryoprotected by immersion in PBS containing 20% sucrose for 3 days at 4 °C, then frozen at −80 °C for slicing. Brains were sectioned through the coronal plane at 10 μm thickness using a cryostat and then mounted on glass slides. Slides were placed in crystal jars with 4% PFA for 15 min and then washed three times with 1X Tris buffer saline supplemented with 0.3% Triton X-100 (TBS-T). Endogenous peroxidases were quenched by immersing sections in a 0.3% hydrogen peroxide methanol solution for 5–10 min, followed by three washes in TBS-T. Then, antibody epitopes were retrieved by placing the slides in a sodium citrate buffer solution (10 mM sodium citrate, 0.05% Tween 20, pH 6.0) and heating to boil. Slides were let to cool down and then washed one time with ddH_2_O to remove the acidity of the citrate buffer, and two more times with TBS-T. Antibody non-specific binding was blocked by incubating brain sections in a blocking solution (1X TBS supplemented with 5% normal goat serum and 0.3% Triton X-100) for 1 h at room temperature. Afterwards, slides were incubated with anti-human p181-Tau mouse monoclonal antibody AT270 (1:500 dilution, cat no. MN1050, Thermo Fisher Scientific). Then, sections were washed three times in TBS-T and incubated for 1 h at room temperature with a 1:200 dilution of Alexa Fluor^TM^ 488 goat anti-mouse secondary antibody (cat. no. A11001, Thermo Fisher Scientific) in blocking solution. Excess secondary antibody was removed by washing slides three times in TBS-T followed by the addition of mounting media supplemented with the nucleic acid stain DAPI (cat. no. P36962, Thermo Fisher Scientific) prior to visualization of sections and image acquisition using the Zeiss LSM 710 confocal laser scanning microscope system (Carl Zeiss Inc., Jena, Germany).

### 2.7. Statistical Analysis 

Outlier values were identified and removed from data sets based on the robust regression and outlier removal test (ROUT) only for C1q in the Hp (2 WT values). No outliers were identified for other analytes. One-way ANOVA was used to determine group differences. Correlation of protein levels between NEVs and brain tissue across mice was assessed calculating the Pearson correlation coefficient, excluding matched samples from subjects providing outlier values (for C1q only).

## 3. Results

### 3.1. Characterization of EVs

NEVs were characterized according to the minimal information for studies of extracellular vesicles 2018 (MISEV2018) guidelines to confirm isolation of nanoparticles containing canonical EV markers [37]. NEV immunoblots showed enrichment for FLOT1, CD9 and CD81, and two neuronal markers, L1CAM and GRIA2 (Figure 1a). Cellular contamination in isolated NEVs was minimal, as little to none Calnexin was detected (Figure 1a). Contamination with albumin was present at a low level (Figure 1a). To further assess NEV purity, we measured levels of two abundant plasma proteins, ApoA and albumin, with ELISAs. ApoA in mice plasma was 111 ± 23 µg/mL and in NEVs 0.198 ± 0.15 µg/mL, indicating ~99% ApoA removal. For albumin 10,354 ± 238 µg/mL were measured in plasma and 151 ± 11 µg/mL in NEV, indicating ~98% removal. 

The simultaneous presence of EV marker CD9 and neuronal marker L1CAM in NEVs was confirmed by FC analysis showing positive fluorescent events, distinct from background noise (from unstained sample) and much more abundant than for EVs immunocaptured using non-specific anti-IgG antibodies (Figure 1b). NTA showed that NEVs/AEVs have a diameter mode of 50–150 nm, consistent with a mixture of exosomes and microvesicles (Figure 1c).

Although the collected plasma volume was not enough for a full characterization of mouse GLAST + AEVs, the demonstrated enrichment of NEVs for canonical EV markers and neuronal markers, the similar range of diameters and concentrations on NTA for NEVs and AEVs, and the fact that our methodology for immunocapture of AEVs from plasma is essentially the same with that for immunocapturing NEVs inspire a certain degree of confidence that AEV isolation was equally successful and resulted in a similar degree of astrocytic-origin enrichment. More broadly, data on enrichment from previous studies using human samples [9,10,16,17] can be reasonably expected to pertain to this study, since the protein epitopes on L1CAM and GLAST targeted for immunoprecipitation in this study are conserved among species.

### 3.2. Total Tau and p181-Tau Levels in NEVs from 3xTg-AD Mice Are Elevated and Correlated with Brain Levels

We sought to confirm that 3xTg-AD mice in our cohort (6 – 10.5 month old) exhibited deposition of human tTau and p181-Tau (known to begin at six months of age, with robust detection at 12 months of age) [25,34]. First, p181-Tau by fluorescence immunohistochemistry was abundant in the Ctx and Hp of a 10.5-month-old 3x-Tg AD mouse compared to WT and 2xTg-AD (human Tau-negative) controls (Figure 2a). Second, Western blot (WB) results from a representative group of 3xTg-AD mice (Figure 2b,c) showed an age-dependent accumulation of tTau and p181-Tau, in the Ctx and Hp of 3xTg-AD compared to WT mice (tTau in Ctx: *p* = 0.0103 at 10 months; in Hp: *p* = 0.0002 at 8 months and *p* = 0.0381 at 10 months; p181-Tau in Ctx: *p* = 0.0214 at 10 months; in Hp: *p* = 0.0077 at eight months and *p* = 0.0099 at 10 months; one-way ANOVA). 

Next, we used a Luminex Multiplex to measure tTau and p181-Tau in NEVs, Ctx and Hp of all 3xTg-AD mice, as well as 2xTg-AD, 5xFAD, and WT mice. As shown in Figure 2d–g, tTau and p181-Tau were increased in the Ctx, Hp and NEVs of 3xTg-AD mice compared to WT, 2xTg-AD and 5xFAD mice (one-way ANOVA; tTau in Ctx: *p* < 0.0001 vs. WT and 2xTg-AD mice, *p* = 0.0019 vs. 5xFAD mice; tTau in Hp: *p* < 0.0001 vs. WT, 2xTg-AD and 5xFAD mice; tTau in NEVs: *p* = 0.0342 vs. WT mice; p181-Tau in Ctx: *p* < 0.0001 vs. WT and 5xFAD, *p* = 0.0102 vs. 2xTg-AD mice; p181-Tau in Hp: *p* < 0.0001 vs. WT and 5xFAD, *p* = 0.0021 vs. 2xTg-AD mice; p181-Tau in NEVs: *p* = 0.0004 vs. WT mice and *p* = 0.0009 vs. 5xFAD mice). Human Tau species in the brain of 3xTg-AD mice increased from 6 to 8 to 10 months of age (Figure 2h,i; tTau in Ctx: *p* = 0.0313 for 8 vs. 10 months; tTau in Hp: *p* = 0.0028 for 6 vs. 10 months and *p* = 0.0082 for 8 vs. 10 months; one-way ANOVA), consistent with immunoblotting results (Figure 2b,c) and previous reports [34,38]. 

We found strong positive correlations for tTau and p181-Tau levels in NEVs and cortical and hippocampal tissues, which were driven by 3xTg-AD mice (Figure 2j–m; tTau: r = 0.3927 with *p* = 0.0092 for NEVs vs. cortex; p181-Tau: r = 0.6718 with *p* < 0.0001 for NEVs vs. Ctx and r = 0.5769 with *p* < 0.0001 for NEVs vs. Hp). 

### 3.3. Aβ42 Levels in NEVs from 5xFAD Mice Are Elevated and Correlated with Brain Levels

Luminex Multiplex results in Figure 3a,b show that 5xFAD mice (~12-month-old) had higher Aβ42 levels in the Ctx, Hp and NEVs compared to 2xTg-AD (~8 month-old), 3xTg-AD (~9 month-old), and WT mice (~9 month-old) (one-way ANOVA; 5xFAD Ctx: *p* = 0.004 vs. WT mice and *p* = 0.0104 vs. 3xTg-AD mice; 5xFAD Hp: *p* < 0.0001 vs. WT and 3xTg-AD mice, and *p* = 0.0132 vs. 2xTg-AD mice; 5xFAD NEVs: *p* = 0.0023 vs. WT mice, *p* = 0.0021 vs. 2xTg-AD mice and *p* < 0.0001 vs. 3xTg-AD mice). NEV and brain Aβ42 levels were strongly positively correlated, with correlations driven by 5xFAD mice (Figure 3c,d; NEVs vs. Ctx, r = 0.6, *p* = 0.001; NEVs vs. Hp, r = 0.7, *p* < 0.0001).

### 3.4. C1q Levels Are Elevated in AEVs of 3xTg-AD Mice and Exhibit a Positive Correlation with Brain Levels

Mouse C1q ELISAs conducted on brain and AEV lysates from 2xTg-AD, 3xTg-AD and 5xFAD mice showed that C1q levels were significantly elevated only in AEVs from 3xTg-AD compared to WT mice, although similar trends were observed for brain lysates from all three AD mouse models (Figure 4a,b; one-way ANOVA; *p* = 0.0049). We found significant positive correlations between C1q in AEVs and cortical and hippocampal tissues across all mouse models (Figure 4c,d; C1q: AEVs vs. Ctx, r = 0.8544, *p* < 0.0001, vs. Hp, r = 0.7051, *p* < 0.0001). 

## 4. Discussion

The purpose of this study was to examine the association between EV biomarkers for AD derived from plasma and their levels in brain tissues, thus demonstrating the potential of using circulating EVs as a window to the brain for the study of AD. To that end, we studied a diverse group of mice (multiple transgenic models for AD and WT, of both sexes and a wide range of ages) to examine whether correlations with circulating EVs exist for a wide range of Aβ, Tau, and complement levels in the brains. 

The study was principally motivated by previous findings showing that NEVs and AEVs from AD patients compared to controls carry high levels of proteins associated with AD pathogenesis, such as tTau, p181-Tau and Aβ42 in NEVs [9,10,12,16,17,19], and complement C1q in AEVs [15]. We previously proposed that Tau and Aβ measured in NEVs provide advantages over their detection in the soluble phase of blood or total plasma EVs including the neuronal source of the EVs and potentially higher levels above conventional detection thresholds [9]. Our group has produced multiple studies of NEV and AEV biomarkers for clinical AD [12,15,16]. Moreover, we have expanded their frame of use to preclinical stages of AD by conducting two large case-control studies: in a study involving longitudinal samples from Baltimore Longitudinal Study of Aging participants cognitively normal at baseline, we demonstrated that a set of NEV biomarkers (including p181-Tau, p-231-Tau and total Tau) was able to predict future AD diagnosis about four years before symptom onset [17]; in a study analyzing longitudinal samples from Wisconsin Registry for Alzheimer’s Prevention participants cognitively normal at baseline, we demonstrated that a set of NEV biomarkers was able to predict future cognitive decline [19]. At this point of development of NEV/AEV biomarkers, a clinicopathologic study based on paired brain-plasma samples from human patients would be required to determine the potential relationship of NEV/AEV biomarkers with AD pathologic stage. As a step towards this direction, we conducted the present study using paired brain-plasma samples from various AD model mice. The results offer support for the use of plasma NEVs biomarkers as surrogates of brain pathology by demonstrating moderately strong correlations between EV biomarkers and levels in matched brain tissue samples across multiple mouse models of amyloidosis and amyloidosis/tauopathy.

Among biomarkers of Tau pathology, we focused on p181-Tau given that it is a well-established marker of neurofibrillary tangles (NFTs) in 3xTg-AD mice [25,34] and AD brains [39], a CSF and plasma biomarker used to support AD diagnosis [40,41,42,43], and the strongest predictor of AD in previous NEV studies [12,17,19]. We have studied mice of different ages ranging from prodromal stage to symptomatic, to better reflect a wide range of pathological severity. Results showed that increased levels of tTau and p181-Tau can be detected in NEVs even in younger 3xTg-AD mice that express mild pathologic changes. Levels of tTau and p181-Tau in NEVs of 3xTg-AD mice are elevated compared to other mice types as early as six months, even prior to obvious development of Tau deposition in the cortex, and remain similarly elevated in older mice with fully developed Tau pathology. These findings are in agreement with our previous findings of elevated Tau in preclinical AD patients [17,19]. While NEV p181-Tau levels were correlated with those in the cortex and hippocampus across all mice, NEV tTau levels were only correlated with levels in the cortex. Given that the accumulations of tTau in the Ctx and Hp of 3xTg-AD mice as they age are similar [25,34], we hypothesize that their disparate correlations with NEV levels could be due to brain region-specific mechanisms underlying the sorting of cargo to and/or secretion of EVs. Alternatively, it could be due to the larger size of the Ctx compared to the Hp, which may produce a larger proportion of circulating NEVs and a stronger biomarker signal. The dynamic ranges of tTau and p181-Tau in 3xTg-AD and WT NEVs were smaller compared to those in the brain (Figure 2d vs. Figure 2e, Figure 2f vs. Figure 2g), which could reflect the overall lower abundance of Tau in NEVs compared to brain lysates. Moreover, a degree of non-specific cross-reactivity of the human tTau and p181-Tau assays with mouse Tau is suggested by low levels in non−3xTg-AD mice (Figure 2b,e–g), consistent with previous observations [44].

Regarding Aβ42 in NEVs as a surrogate of Aβ brain accumulation, we expected age-dependent differences in NEVs between mice types consistent with the differential appearance of Aβ pathology in the brain and its age-dependent effects on other pathogenic processes [45] (specifically, 5xFAD mice exhibit an earlier onset of Aβ accumulation, at 2 months of age [29], compared to 2xTg-AD mice, at seven months of age [28,46], and 3xTg-AD mice, which show a trend at 5-7-months and reach full loads at 13 months of age [25,34,47]). Consistently, we found that the 12-month-old 5xFAD mice had increased Aβ42 levels in the cortex, hippocampus and NEVs compared to the younger 2xTg-AD and 3xTg-AD, as well as WT mice (Figure 3). The level of Aβ42 in NEVs was positively correlated with its levels in the Ctx and Hp across mice. Importantly, similar to previous studies, the Aβ levels in EVs were relatively low and the Aβ42/Aβ40 ratio was higher [48].

The presence of Tau and Aβ in circulating NEVs highlights the potential role of EVs in the spreading of Tau and Aβ pathologies in AD that has been investigated in several studies. For example, total EVs isolated from the brains of the tauopathy mouse model rTg4510 carry high levels of phosphorylated Tau and can induce Tau aggregation in recipient cells in vitro [49] and in vivo [50]. Moreover, inhibition of EV secretion in 5xFAD mice results in reduced Aβ plaque burden [51], whereas EVs isolated from the CSF and brains of AD patients carry increased Aβ42 and mediate neurotoxicity in vitro that can be blocked by anti-Aβ antibodies [48,52]. Moreover, the presence of Tau and Aβ in circulating NEVs suggest a potential role in reducing CNS burden by clearing them to peripheral blood. While this study establishes the connection between peripheral EV and brain tissue levels, additional research is needed to uncover the physiological and pathological roles of NEVs and AEVs in the periphery.

Regarding complement, we previously reported that AEVs of AD patients carry increased levels of its components, with major differences shown for C1q, compared to controls [15]. Moreover, increased complement has been found in the brain and CSF of AD patients [53,54]. We recently demonstrated that AEVs and NEVs of AD patients can exert complement-mediated neurotoxicity to recipient neurons [55]. A recently proposed mechanism holds that complement-mediated synaptic pruning that is dormant in the adult brain becomes active in AD and mediates synaptic loss, with accumulation of oligomeric Aβ and hyperphosphorylated Tau leading to astrocyte overexpression of complement C1q and C3, which prime synapses for microglial phagocytosis [30,54,56,57]. To our knowledge, our study is the first to evaluate C1q levels in 5xFAD and 3xTg-AD mice, or in AEVs in any mouse model of AD. We observed increased levels of C1q in AEVs from 3xTg-AD mice compared to WT controls and similar trends for other AD models, perhaps suggesting that the combination of amyloidosis and tauopathy more potently drives neuroinflammation. Moreover, it has been previously reported that 3xTg mice manifest more neuronal death than other AD models [25], which can be a cause or a consequence of the observed elevated complement. Finally, we showed that AEV and brain levels of C1q are strongly correlated with each other (Figure 4). 

We have previously demonstrated that classic AD pathogenic proteins (including total Tau, p181Tau, and Aβ42) are elevated not only in NEVs, but also in AEVs of AD patients [16]. Therefore, it would be of interest to examine whether their levels in AEVs are associated with brain levels. Whereas limited sample availability dictated our choice to quantify these biomarkers only in NEVs, future studies should assess a wider array of candidate EV biomarkers in both NEVs and AEVs.

This study focused on comparing circulating NEV/AEV biomarkers against tissue levels. This way we sought to connect our EV biomarker findings with the large body of literature that have used tissue levels of total Tau, p181Tau, and Aβ42 in AD mouse models as surrogates for brain pathologies. In recent years various methods have been developed for isolating EVs from brain interstitial fluid [58,59], although no consensus currently exists regarding the optimum methodology. As this body of literature grows and extends to AD mouse models, future studies may compare biomarker levels in circulating NEVs/AEVs and brain EVs. Moreover, another potential comparator of circulating NEVs/AEVs is CSF EVs; however, sampling adequate CSF from the narrow ventricles of the mouse brain is challenging and carries the almost inescapable risk of blood contamination of CSF during its acquisition. A less invasive gradual CSF acquisition using chronic indwelling implants may solve this problem and offer the basis for a comparison between circulating NEVs/AEVs and CSF EVs in the future.

Altogether, our findings provide strong support for using plasma NEVs and AEVs as a source of biomarkers to probe diverse mechanisms involved in AD. We found variability in levels of pathologic molecules in the brains and plasma NEVs and AEVs, even among genetically similar mice, likely due to variability in age, sex and other unaccounted for biological factors. Remarkably, the variability in brain levels was reflected on EVs, further suggesting that the latter may be used as a precision medicine tool, as previously proposed [60]. This possibility is further supported by human NEV studies showing remarkably low within individual variability over months [61] and years [17] suggesting the presence of stable EV signatures in peripheral blood. To further explore this possibility, future clinicopathological studies may assess the relationships between EV biomarkers, brain levels of pathogenic molecules, and measures of neuropathology in the brains of AD patients. 

## 5. Conclusions

We showed that levels of Aβ and Tau in plasma NEVs and C1q in plasma AEVs moderately-strongly reflect their levels in the brain. These results strongly support the validity of circulating EV biomarkers as surrogates of brain pathology and their potential use as a form of “liquid biopsy” for enabling AD precision medicine. EVs may help us unravel the temporal and etiologic sequence of multiple pathogenic processes in neurodegenerative diseases by enabling the study of their evolution over time in living humans. Finally, the demonstration of strong correlations between proteins with pathogenic potential (e.g., C1q) in plasma EVs and brain tissues strengthen the rationale for targeting them therapeutically or even help identify novel therapeutic targets.

## Figures and Tables

**Figure 1 cells-10-00993-f001:**
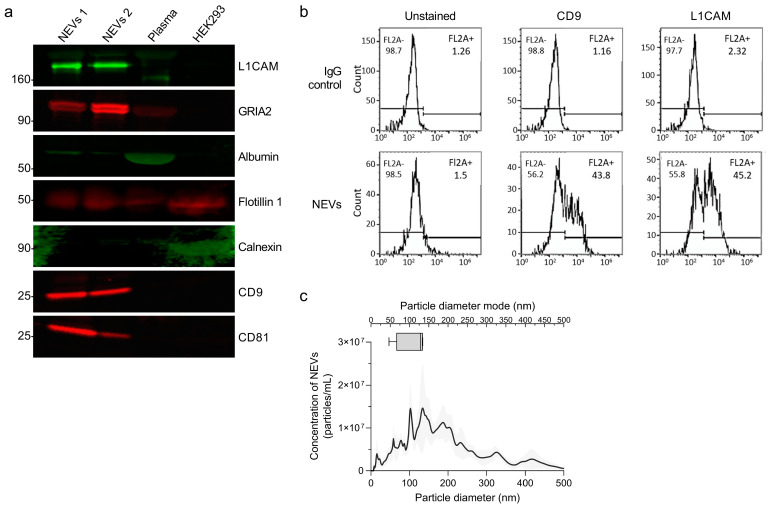
Characterization of extracellular vesicles. (**a**) WBs showing the protein levels of the intra-vesicular EV markers Flotillin 1, and transmembrane EV markers CD9, CD63 and CD81, used as positive EV markers, the neuronal specific markers L1CAM and GRIA2 as indicators of neuronal enrichment, the endoplasmic reticulum marker Calnexin used as a negative EV marker, and albumin as an indicator of peripheral contaminants co-precipitated with EVs, in 20 µg of NEVs from two WT mice (NEVs 1 and NEVs 2) and 1:50 diluted neat plasma, as well as in 20 µg of HEK293 cell lysate used as a negative control for the enrichment of neuronal- and EV-specific markers. (**b**) FC analysis of NEVs and non-specific EVs immunocaptured with anti-IgG antibodies fluorescently labelled with antibodies against the EV marker CD9 and the neuron specific marker L1CAM. The fluorescent signal from unstained EVs was used to establish the fluorescence threshold. (**c**) The graph presents EV concentration (particles/EVs per mL) as a function of particle diameter (determined using NTA) for immunoprecipitated NEVs isolated from the plasma of 4 WT mice in duplicate. In the upper part, a box and whiskers graph plots the particle diameter mode range.

**Figure 2 cells-10-00993-f002:**
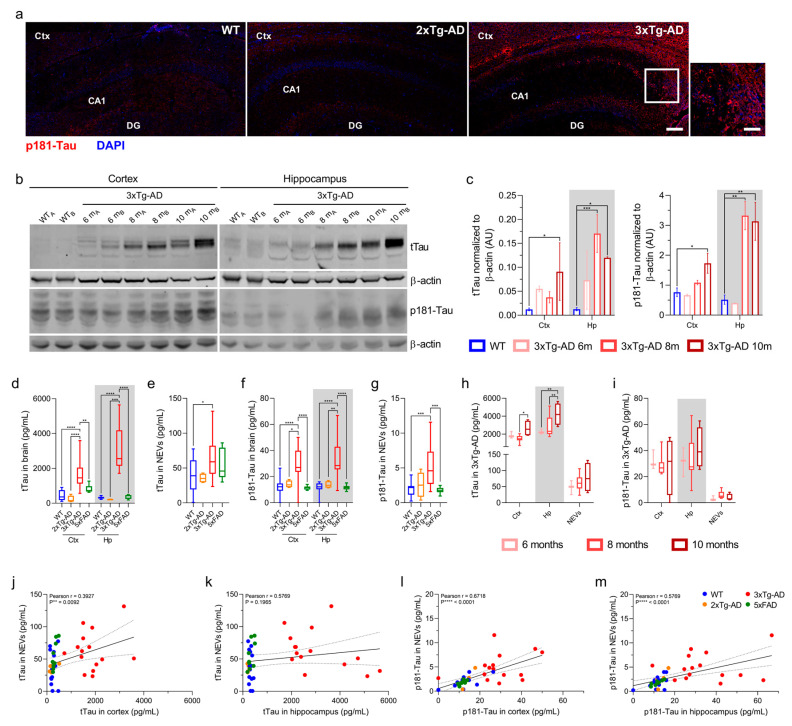
tTau and p181-Tau in NEVs are positively correlated with brain levels. (**a**) p181-Tau fluorescence immunohistochemistry (red), with DAPI-stained nuclei (blue), of the Ctx and Hp of a 10.5-month-old 3xTg-AD mouse compared to WT (5.7 months) and 2xTg-AD (8.3 months) negative controls (20_X_, scale bar, 200 µm). Square inset in 3xTg-AD image: scale bar, 100 µm. (**b**) tTau (HT7) and p181-Tau (AT270) WB, resolved in individual membranes, showing brain levels of two 3xTg-AD mice per age group compared to WT (6-month-old). (**c**) WB quantification of tTau and p181-Tau normalized to β-actin (one-way ANOVA; tTau in Ctx: * *p* = 0.0103, 10 months; tTau in Hp: *** *p* = 0.0002, 8 months, and * *p* = 0.0381, 10 months; p181-Tau in Ctx: * *p* = 0.0214, 10 months; p181-Tau in Hp: ** *p* = 0.0077, 8 months, and ** *p* = 0.0099, 10 months). Mice used for tTau assessment: WT, n = 6, age mean = 5.3 months; 3xTg-AD, two of which are 6-months, six of which are 8-months and two of which are 10-months old (as indicated by age-specific colored dots). Mice used for the p181-Tau assessment: n = 2 for all groups. (**d**–**g**) Box-plots displaying human tTau (**d**,**e**) and p181-Tau (**f**,**g**) levels in brains and NEVs as measured by a Luminex-multiarray (one-way ANOVA; tTau in Ctx: **** *p* < 0.0001 vs. WT and 2xTg-AD, *** *p* = 0.0019 vs. 5xFAD; tTau in Hp: **** *p* < 0.0001 vs. WT, 2xTg-AD and 5xFAD; tTau in NEVs: * *p* = 0.0342 vs. WT; p181-Tau in Ctx: **** *p* < 0.0001 vs. WT and 5xFAD, * *p* = 0.0102 vs. 2xTg-AD; p181-Tau in Hp: **** *p* < 0.0001 vs. WT and 5xFAD, ** *p* = 0.0021 vs. 2xTg-AD; p181-Tau in NEVs: *** *p* = 0.0004 vs. WT and *** *p* = 0.0009 vs. 5xFAD). No outliers were identified. (**h**,**i**) tTau (**h**) and p181-Tau (**i**) Luminex-multiarray results in 3xTg-AD mice organized by age groups (one-way ANOVA; tTau in Ctx: * *p* = 0.0313 for 8 vs. 10 months; tTau in Hp: ** *p* = 0.0028 for 6 vs. 10 months and * *p* = 0.0082 for 8 vs. 10 months). (**j**–**m**) tTau (**j**,**k**) and p181-Tau (**l**,**m**) levels in NEVs in relation to Ctx and Hp. Pearson correlation coefficient shown for each graph.

**Figure 3 cells-10-00993-f003:**
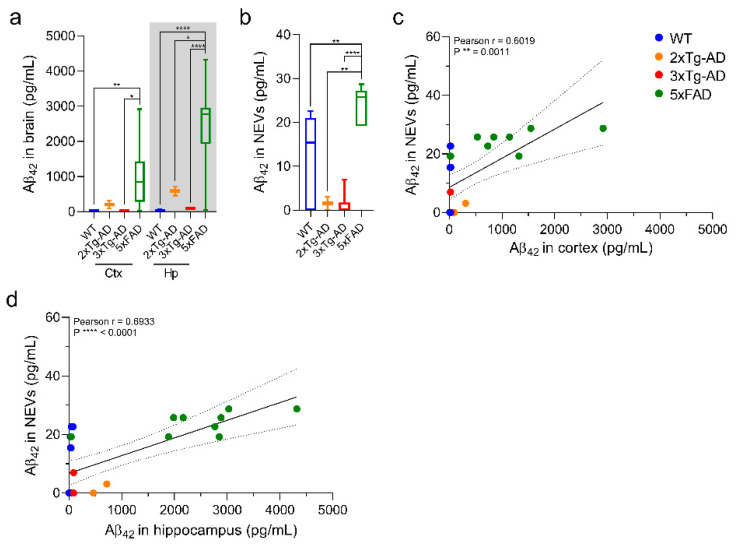
Aβ42 levels in NEVs are positively correlated with brain levels. (**a**,**b**) Box-plots displaying human Aβ42 protein levels in brains (**a**) and NEVs (**b**) of different AD mouse models and WT mice as measured by a Luminex-multiarray. Results show significantly increased levels of Aβ42 in the brains and NEVs of 5xFAD mice compared to WT, 2xTg-AD and 3xTg-AD mice (one-way ANOVA; Ctx: ** *p* = 0.004 vs. WT mice and * *p* = 0.0104 vs. 3xTg-AD mice; Hp: **** *p* < 0.0001 vs. WT and 3xTg-AD mice, and * *p* = 0.0132 vs. 2xTg-AD mice; NEVs: ** *p* = 0.0023 vs. WT mice, ** *p* = 0.0021 vs. 2xTg-AD mice and **** *p* < 0.0001 vs. 3xTg-AD mice). N: 10 WT, 2 2xTg-AD, 6 3xTg-AD and 9 5XFAD. (**c**,**d**) Luminex-multiarray Aβ42 levels in NEVs in relation to Ctx (**c**) and Hp (**d**) show statistically significant positive correlations as indicated by the Pearson correlation coefficient shown in each graph. All mice indicated in the methods section were included in the analysis and no outliers were identified.

**Figure 4 cells-10-00993-f004:**
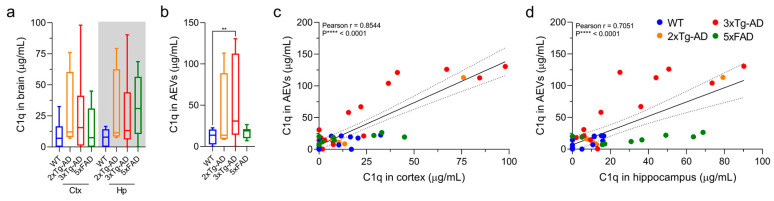
Brain and AEV C1q are increased in 3xTg-AD mice and positively correlate across mouse groups. (**a**,**b**) Box plots displaying complement C1q (**a**,**b**) protein levels in brains and AEVs of different AD mouse models and WT mice as measured by mouse C1q ELISA. Results in (**b**) show significantly increased levels of C1q in AEVs of 3xTg-AD mice compared to WT controls (one-way ANOVA; ** *p* = 0.0049). (**c**,**d**) ELISA C1q levels in AEVs in relation to Ctx and Hp show statistically significant positive correlations as indicated by the Pearson correlation coefficient shown in each graph. All mice indicated in the methods section were included in the analysis. Several outliers were identified and removed based on the ROUT test (hippocampal C1q: 2 WT values). Matched samples from subjects providing outlier values were removed for the correlation analysis.

## Data Availability

The datasets generated during the current study are available from the corresponding author upon reasonable request.

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
