# Peer review of "Neuronal and Astrocytic Extracellular Vesicle Biomarkers in Blood Reflect Brain Pathology in Mouse Models of Alzheimer’s Disease"

_cells, 2021, doi:10.3390/cells10050993_

Round 1

Reviewer 1 Report

In recent years, the extracellular vesicle (EV) field has vastly expanded, with findings providing functional implications for both normal physiological and pathological states.

The manuscript shows as Neuronal and astrocytic extracellular vesicle biomarkers in blood reflect brain pathology in mouse models of Alzheimer's disease, the findings provide strong support for using plasma NEVs and AEVs as a source of biomarkers to probe diverse mechanisms involved in AD and potentially open the door to precision therapeutic approach.

Circulating neuronal extracellular vesicles (NEVs) and astrocytic EVs (AEVs) are the key cores of the paper.

Heterogeneity of EV populations has become an increasingly intense area of focus since it would be highly desirable to identify biomarkers for EVs released from a specific cell population. However, the large variety of EVs with varying characteristics brings into question what processes can actually be precisely identified in a complex cellular environment, especially given that heterogeneity of EVs exists not only across parent cell populations but also as released from a single-cell type.

  1. So, I strong suggest the author show the pictures of Size exclusion chromatography of the NEVs and AEVs. Size exclusion chromatography enables size-based separation on a single column, with the majority of EVs eluting before soluble components such as proteins and HDL.
  2. EV cargo quantification of NEVs and AEVs are needed.
  3. Methods to Measure Single EVs, EM is the gold standard method for imaging EVs. The resolution of EM images is ≈1 to 3 nm for transmission electron microscopy and ≈5 nm for EVs detection by scanning EM. Here, I suggest the author will focus on transmission electron microscopy, which covers most EM studies on EVs. The TEM images of NEVs and AEVs are needed.

Various approaches have been used to assess and characterize EVs selectively derived from cells in the brain. For instance, CSF has been collected and analyzed to examine EVs localized within the intracerebral space. However, it is often challenging to obtain a pure sample because techniques to access the interstitial fluid of the brain or spinal cord can induce puncture of blood vesicles, resulting in blood-derived EV contamination. To overcome this issue, chronic indwelling implants may be used to collect small samples over time.

So, the author also need address it is advantage of their method in the discussion section.

I feel the overall manuscript has potential and has interesting for readers, I would be happy to review the revised manuscript.

Reviewer 2 Report

The Article entitled “Neuronal and astrocytic extracellular vesicle biomarkers in blood reflect brain pathology in mouse models of Alzheimer's disease” is well structured and organized manuscript. I have concerns regarding detection of biomarkers and selection of biomarkers.

  • Authors selected only 3 biomakers 2 for NEVs (total pTAU, p181) and 1 for AEVs (C1q) in the plasma. It would be interesting pTAU in AEVs too. Also the investigation of expression of well as C1q in NEV would be interesting
  • Data for astrocyte-specific proteins such as GFAP, excitatory amino acid transporter 1 (SLC1A3/GLAST) and glucose transporter member 1 (SLC2A1/GLUT1) should be shown in the AEVs
  • Evaluation of endogenous complement regulatory proteins like CD59 in AEVs and NEVs would be of interest.
  • The expression of pTAU181 by histology is not convincing. Picture quality should be improved
  • In western blot the expression of beta action in hippocampus and cortex tissues was different. Also with in hippocampus the expression of beta actin is varying between samples. Authors must ensure the validity of the WB testing.

Reviewer 3 Report

The relationship of EV biomarkers and brain pathology is not clear, and thus the study from Delgado-Peraza et al. is of high interest. Demonstrating a relationship between levels of pathological proteins accumulating in the brains and EVs of AD mouse models would be a significant contribution towards establishing AD disease biomarkers. Here the authors showed in several AD mouse models, across a range of ages, that the pathological proteins most closely associated with the AD models were strongly correlated between EVs and paired brain Ctx and Hp. The authors are highly experienced in performing the technically challenging enrichment of specific populations of EVs, and overall presented convincing data showing quality controls for EV purity, composition, and size, and used reliable experimental approaches for quantification. The results are of high quality and the conclusions are nicely supported by the data.

Primary comment:

The results feel somewhat incremental towards establishing a validated biomarker candidate. Given the robust methods developed and validated for quantification of proteins associated with AD in EVs, it would significantly elevate the manuscript if the authors could more strongly link one of the most highly correlated proteins to disease severity or stage, or alternatively, could demonstrate whether one of the candidates could be used in a predictive capacity.  The authors hinted at these issues when they briefly noted the variability of absolute levels between genetically identical mice, and mentioned the possibility of age as a factor, so this is likely a challenging aspects that could be outside the scope of this study; however, if the authors could provide more context to these possibilities this would also be a useful addition to the manuscript.

Author Response

RESPONSE TO REVIEWER 2 COMMENTS

The relationship of EV biomarkers and brain pathology is not clear, and thus the study from Delgado-Peraza et al. is of high interest. Demonstrating a relationship between levels of pathological proteins accumulating in the brains and EVs of AD mouse models would be a significant contribution towards establishing AD disease biomarkers. Here the authors showed in several AD mouse models, across a range of ages, that the pathological proteins most closely associated with the AD models were strongly correlated between EVs and paired brain Ctx and Hp. The authors are highly experienced in performing the technically challenging enrichment of specific populations of EVs, and overall presented convincing data showing quality controls for EV purity, composition, and size, and used reliable experimental approaches for quantification. The results are of high quality and the conclusions are nicely supported by the data.

Point 1. The results feel somewhat incremental towards establishing a validated biomarker candidate. Given the robust methods developed and validated for quantification of proteins associated with AD in EVs, it would significantly elevate the manuscript if the authors could more strongly link one of the most highly correlated proteins to disease severity or stage, or alternatively, could demonstrate whether one of the candidates could be used in a predictive capacity. The authors hinted at these issues when they briefly noted the variability of absolute levels between genetically identical mice, and mentioned the possibility of age as a factor, so this is likely a challenging aspect that could be outside the scope of this study; however, if the authors could provide more context to these possibilities this would also be a useful addition to the manuscript.

Response 1. Our group has produced multiple studies of blood neuronal and astrocytic-enriched extracellular vesicle (EV) biomarkers for Alzheimer’s disease (AD) at the clinical stage 1-3. Moreover, we have expanded their frame of use to preclinical stages of AD by conducting two large case-control studies: in a study examining longitudinal samples from Baltimore Longitudinal Study of Aging participants cognitively normal at baseline, we demonstrated that a set of NEV biomarkers (including p181-Tau, p-231-Tau and total Tau), were able to predict future AD diagnosis about 4 years before symptom onset  4; in a study examining longitudinal samples from the Wisconsin Registry for Alzheimer’s Prevention participants cognitively normal at baseline, we demonstrated that a set of NEV biomarkers were able to predict future cognitive decline 5.

As the reviewer implies, a clinicopathologic study determining the potential relationship between EV biomarkers and AD pathologic stage would be an interesting and important investigation, which would require paired brain-plasma samples from human patients. It is certainly our intent to conduct such a study in the future. As a step towards this direction, we conducted the present study using paired brain-plasma samples from various AD model mice. The results show that increased levels of tTau and p181-Tau can be detected in NEVs and brains of 3xTg-AD mice even in younger mice that represent a prodromal phase of brain pathology. To illustrate this point we identify these younger 3xTg-AD mice even in Figure 2j-m with a different color and show their overlap with older 3xTg-AD mice. The obvious conclusion is that levels of tTau and p181-Tau in NEVs of 3xTg-AD mice are elevated compared to other mice types even as early as 6 months, prior to obvious development of Tau deposition in the cortex, and remain similarly elevated in older mice with fully developed Tau pathology. These findings are in perfect agreement with our previous findings of elevated Tau in presymptomatic AD patients.

We now state in the discussion: “Our group has produced multiple studies of NEV and AEV biomarkers for clinical AD1-3. Moreover, we have expanded their frame of use to preclinical stages of AD by conducting two large case-control studies: in a study involving longitudinal samples from Baltimore Longitudinal Study of Aging participants cognitively normal at baseline, we demonstrated that a set of NEV biomarkers (including p181-Tau, p-231-Tau and total Tau) was able to predict future AD diagnosis about 4 years before symptom onset 4; in a study analyzing longitudinal samples from Wisconsin Registry for Alzheimer’s Prevention participants cognitively normal at baseline, we demonstrated that a set of NEV biomarkers was able to predict future cognitive decline 5. At this point of development of NEV/AEV biomarkers, a clinicopathologic study based on paired brain-plasma samples from human patients would be required to determine the potential relationship of NEV/AEV biomarkers with AD pathologic stage. As a step towards this direction, we conducted the present study using paired brain-plasma samples from various AD model mice. The results show that increased levels of tTau and p181-Tau can be detected in NEVs and brains of 3xTg-AD mice even in younger mice that represent a prodromal phase of brain pathology (highlighted in Figure x with y color). Levels of tTau and p181-Tau in NEVs of 3xTg-AD mice are elevated compared to other mice types even as early as 6 months, even prior to obvious development of Tau deposition in the cortex, and remain similarly elevated in older mice with fully developed Tau pathology. These findings are in perfect agreement with our previous findings of elevated Tau in presymptomatic AD patients. Moreover, we observed statistical trends for positive correlations between NEV levels and mouse age for tTau (Figure 2h) and p181-Tau (Figure 2i), respectively, further supporting the value of following Tau in plasma NEVs as a predictor of disease severity.” (lines 426-449)

REFERENCES

1          Fiandaca, M. S. et al. Identification of preclinical Alzheimer's disease by a profile of pathogenic proteins in neurally derived blood exosomes: A case-control study. Alzheimers Dement 11, 600-607 e601, doi:10.1016/j.jalz.2014.06.008 (2015).

2          Goetzl, E. J. et al. Cargo proteins of plasma astrocyte-derived exosomes in Alzheimer's disease. FASEB J 30, 3853-3859, doi:10.1096/fj.201600756R (2016).

3          Goetzl, E. J., Schwartz, J. B., Abner, E. L., Jicha, G. A. & Kapogiannis, D. High complement levels in astrocyte-derived exosomes of Alzheimer disease. Annals of neurology 83, 544-552, doi:10.1002/ana.25172 (2018).

4          Kapogiannis, D. et al. Association of Extracellular Vesicle Biomarkers With Alzheimer Disease in the Baltimore Longitudinal Study of Aging. JAMA Neurol, doi:10.1001/jamaneurol.2019.2462 (2019).

5          Eren, E. et al. Extracellular vesicle biomarkers of Alzheimer's disease associated with sub-clinical cognitive decline in late middle age. Alzheimers Dement 16, 1293-1304, doi:10.1002/alz.12130 (2020).

Round 2

Reviewer 1 Report

All questions were answered quite well in the response letter. I totally agree now.

Reviewer 2 Report

Authors attempt to address the clarifications of the reviewers. The manuscript is in acceptable format now.